# Sex-Biased Expression of Pharmacogenes across Human Tissues

**DOI:** 10.3390/biom11081206

**Published:** 2021-08-13

**Authors:** Maria Laura Idda, Ilaria Campesi, Giovanni Fiorito, Andrea Vecchietti, Silvana Anna Maria Urru, Maria Giuliana Solinas, Flavia Franconi, Matteo Floris

**Affiliations:** 1Institute of Genetics and Biomedical research, 07100 Sassari, Italy; marialaura.idda@irgb.cnr.it; 2Department of Biomedical Sciences, University of Sassari, 07100 Sassari, Italy; icampesi@uniss.it (I.C.); gfiorito@uniss.it (G.F.); andrea.vecchietti@hotmail.it (A.V.); gsolinas@uniss.it (M.G.S.); 3Unit of Environmental Epidemiology, School of Public Health, Imperial College, London SW7 2AZ, UK; 4Hospital Pharmacy Unit, Trento General Hospital, Autonomous Province of Trento, 38122 Trento, Italy; silvanaurru@gmail.com; 5Department of Chemistry and Pharmacy, School of Hospital Pharmacy, University of Sassari, 07100 Sassari, Italy; 6National Laboratory of Pharmacology and Gender medicine, National Institute of Biostructure and Biosystems, 00136 Rome, Italy; franconi@uniss.it

**Keywords:** pharmacogenes, transcripts, sex-bias, drug metabolism, sex differences

## Abstract

Individual response to drugs is highly variable and largely influenced by genetic variants and gene-expression profiles. In addition, it has been shown that response to drugs is strongly sex-dependent, both in terms of efficacy and toxicity. To expand current knowledge on sex differences in the expression of genes relevant for drug response, we generated a catalogue of differentially expressed human transcripts encoded by 289 genes in 41 human tissues from 838 adult individuals of the Genotype-Tissue Expression project (GTEx, v8 release) and focused our analysis on relevant transcripts implicated in drug response. We detected significant sex-differentiated expression of 99 transcripts encoded by 59 genes in the tissues most relevant for human pharmacology (liver, lung, kidney, small intestine terminal ileum, skin not sun-exposed, and whole blood). Among them, as expected, we confirmed significant differences in the expression of transcripts encoded by the cytochromes in the liver, CYP2B6, CYP3A7, CYP3A5, and CYP1A1. Our systematic investigation on differences between male and female in the expression of drug response-related genes, reinforce the need to overcome the sex bias of clinical trials.

## 1. Introduction

The individual response to drugs is a complex mechanism finely regulated by several factors: personal genetic background, environmental influences (exposure to toxins, diet, and smoking), other personal characteristics (age, sex, body size, and ethnicity), and disease (liver and renal pathological states, diabetes, and obesity) [1]. Genetic variations have been estimated to contribute between 20–30% to variability in response to drugs, and the identification and characterization of pharmacogenetic variants in diverse populations is still an ongoing attempt [2]. Among the other factors, the sex of the individual can drastically influence the response to drugs. Females and males can react differently to the same therapeutic regimen due to sex-specific variances in pharmacokinetics (ADMETox) and pharmacodynamics profiles, which very often originate from physiological differences between the genders [3,4,5,6]. For example, the expression and activity of drug-metabolizing CYP450 enzymes can be affected by many factors—including genetic polymorphisms and sex—leading to changes in the metabolism of drugs and their therapeutic effect [7]. Notably, women consume more drugs than men (https://ec.europa.eu/eurostat/data/database, accessed on 13 August 2021). This could depend on multiple factors such as a higher number of visits that they make to their doctor [8]. Besides, drug consumption prevails in old age, and the majority of elderly are women, who are also affected by chronic diseases more than men are [9]. The greater use of drugs, the low enrollment of women in clinical trials (“gender bias”), together with of above-reported sex differences in genes expression, could lead to a greater probability of running into adverse drug reactions (ADR).

Generally speaking, there are really few sex-specific dosage recommendations for almost all prescribed drugs. One example is represented by the dosage of zolpidem, a nonbenzodiazepine hypnotic drug. To decrease the risk of ADR in women, the U.S. Food and Drug Administration (FDA) recommended a 50% lower dosage of zolpidem in women (https://www.fda.gov/drugs/drug-safety-and-availability/fda-drug-safety-communication-fda-approves-new-label-changes-and-dosing-zolpidem-products-and, accessed on 13 August 2021). Men metabolize at a double rate the 10 mg dosage of zolpidem than women [10,11]. Moreover, men have a lower plasma concentration of zolpidem and a shorter clearance than women. In particular, after sublingual zolpidem, the peak concentration is 45% higher in women than men [12].

Sex differences involve almost all ADMETox parameters [13]: variation in the expression level and activity of genes involved in drug processing and action (‘pharmacogenes’) can affect drug response and toxicity, especially in tissues of pharmacological importance; nevertheless, biological factors and mechanisms that regulate sex differences are poorly studied and understood. In this regard, it is relevant to underline that both exogenous (oral contraceptives, hormonal replacement therapy, etc.) and endogenous sex hormones and their variations in activities (menstrual cycle, pregnancy, lactation, etc.) may lead to differences in response to drugs [14]. For example, the metabolism of proguanil, a drug used to treat and prevent malaria, is reduced by oral contraceptives and pregnancy [15]. Importantly, the basal activity of CYP2D6, characterized by a marked polymorphism, which can indicate different activities among individuals [16], is not sex divergent, but it is upregulated by pregnancy, oral contraceptives, and hormone replacement therapy [17,18]. As antipsychotics (risperidone, thioridazine, perphenazine, fluphenazine, zuclopenthixol, haloperidol, and chlorpromazine) are a substrate of the CYP2D6 enzyme, a specific dose during pregnancy should be indicated [19]. Finally, it is well known that some diseases such as diabetes mellitus and COVID-19 can modify drug metabolism either through glycosylation [20] or variation of activity through inflammatory cytokines [21]. Unfortunately, it is still not known if it occurs in a sex-specific manner.

Here, we generated a catalogue of transcripts differentially expressed in either sex to identify candidate pharmacogenes (genes of pharmacological importance) and mechanisms explaining sex-specific responses to drugs. To this end, we used data from Genotype-Tissue Expression project (GTEx, v8 release), a database of transcriptomics studies performed on 838 adult individuals in 44 different tissues, and decided to focus our analysis on 6 tissues, the most relevant for the pharmacokinetics of existing drugs. We then concentrated on all the pharmacogenes defined as enzymes, transporters, carriers, and targets by DrugBank. We found sex-differentiated expression of 99 transcripts encoded by 59 genes implicated in pharmacological-ADMETox, of which 6 are very important pharmacological (VIP) genes. As expected, differential expression in the cytochrome P450 family was identified in several tissues, including the liver and whole blood. Our results highlight relevant sex differences in tissue-specific expression of transcripts encoded by pharmacogenes. Furthermore, it reinforces the urgent need to overcome sex bias in clinical trials and—most importantly—confirms the need to consider sex-specific dosing recommendations for a large number of prescribed drugs.

## 2. Materials and Methods

### 2.1. Data Sources

RNA-seq transcript read counts and de-identified sample annotations were downloaded from the GTEx project (v8 data release) [22]. Only files with read counts from 6 organs (liver, lung, renal cortex, small intestine, skin, and whole blood) were considered for further analysis.

### 2.2. Statistical Methods

Sex-differential expression was investigated using the DESeq2 Bioconductor package within the R statistical environment [23]. Briefly, DESeq2 identify differentially expressed genes through a multistep approach: (i) computation of the normalization factors for each sample to adjust for possible batch effect; (ii) estimation of per-transcript dispersions through a weighted local regression of dispersions over base means on the logarithmic scale (iii) fit a generalized linear model (GLM), under the assumption of a negative binomial distribution of RNA-counts per transcript, (iv) calculation of the Wald test statistics to identify differentially expressed transcripts between male and female. Transcripts with average read counts <10 were excluded from subsequent analysis. In Table 1, we reported the number of transcripts and sample characteristics description for each tissue.

We identified transcripts differentially expressed between males and females through a transcriptome-wide analysis (DESeq2 GLM model), using RNA counts as the dependent variable and gender as the predictor adjusting for chronological age as a covariate. To take into account possible statistical confounding introduced by batch effect and cell type heterogeneity, we used a reference-free algorithm to compute surrogate variables (SVs), implemented in the R package sva [24]. The optimal number of SVs was computed according to the Leek method [24], and finally SVs were included in the regression model as additional covariates. For each transcript, the effect size was expressed as the base 2 logarithm of the fold change (log2FC). We considered men as the reference group, with positive values of log2FC indicating genes overexpressed in females compared to men and vice versa: that is, a positive log2FC indicates overexpression in females and negative log2FC indicates overexpression in men. All analyses were adjusted for multiple comparisons using the Benjamini–Hochberg false discovery rate (FDR). Here, we considered as statistically significant all the genes with FDR q-value lower than 0.05 and FC lower than 0.6 or higher than 1.4 (corresponding to at least 40% differences between male and female). We focused our subsequent analysis on transcripts expressed by genes with a role in drug response. In more detail, we compiled a comprehensive list of 3984 pharmacologically relevant genes from two authoritative and freely available web resources, PharmGKB [25] and DrugBank [26].

A recent study investigated sex-specific gene expression on the same dataset we used but with a slightly different statistical approach [27]. Specifically, Oliva et al. identified sex-specific gene expression using a two-steps approach: First, they ran a tissue-specific regression model, and then a meta-analysis across different tissues. Such a procedure prioritizes sex-specific genes in which the effect on gene expression is common across tissues while penalizes genes in which differential effect of gene expression is tissue-specific. Instead, we focused our investigation on drug-related tissues only, analyzing each tissue independently. In addition, we defined a more stringent threshold to define significant associations based on FDR q-value < 0.05 and FC lower than 0.6 or higher than 1.4. On the contrary, Oliva et al. did not consider any threshold based on FC. Finally, in both papers, we used surrogate variables (SVs) analysis to account for cell type heterogeneity, but using different analytical packages, which may produce slightly different results when including SVs as adjustments in the regression models. Despite such analytical differences, more than 95% of the sex-specific genes we identified are described as differentially expressed by gender in Oliva et al. only, making us confident about the robustness of the results.

### 2.3. Criteria for Pharmacogene Inclusion

Very important pharmacogene (VIP) provided by PharmGKB curation and classification according to DrugBank are described in Table 2.

## 3. Results

### 3.1. Sex Effects on Drug Response (SBDR) Genes

To identify sex effects on gene expression of pharmacogenes, public data from GTEx project v8 data release were used and sex-biased drug response (SBDR) gene expression was calculated in all the 44 tissues present in GTEX. Sex-biased gene expression was quantified in each of the tissue sources for all genes expressed in at least one tissue and 35,341 transcripts in total were considered for further analysis. For each tissue, a linear model—which considers sample and donor characteristics to identify sex-biased gene expression that does not come from differences due to sample composition and cell type abundances—was applied. We focused on genes relevant for ADMETox that belong to one of the relevant classes defined by PharmGKB (VIP, very important pharmacogenes) and DrugBank (drug carrier, transporter, enzyme, target) [25,26].

In the 44 tissues, we identified a total of 1854 transcripts from 756 SBDR genes (FDR ≤ 0.05), with 28.3% (759/2,687) differentially expressed in at least one of the analyzed tissues (p.adj < 0.05) (Appendix A). Subsequently, the most relevant tissues implicated in ADMETox (liver, kidney, small intestine, skin, and whole blood) were deeply investigated.

The analysis focused on genes, which have at least 40% of up or downregulation in females compare to males and end up with the identification of 452 transcript genes with 1 to 91 different transcripts discovered per tissue (Figure 1A). Interestingly, the highest number of SBDR transcripts is present in the thyroid, of which 90% belong to DrugBank targets and only 3% are defined as VIP. While the lower amount of SBDR transcript is present in kidney cortex and breast mammary tissue, with ≤3 SBDR transcripts. For all the genes considered in this analysis, 8% belong to the VIP class. Concerning the classification of DrugBank, 88% of the genes are targets, 37 are enzymes, 11 transporters, and only 4 carriers (Figure 1B,C).

### 3.2. Effects of SBDR in 6 Tissues Most Relevant for Drug Pharmacokinetics

Sex differences in the human transcriptome across the 6 main tissues implicated in drug metabolism were subsequently characterized (liver, lung, kidney, small intestine *terminal ileum*, skin not sun-exposed, and whole blood), using data from 838 individuals in total (557 males, 281 females) (Figure 2A). Volcano plots were generated to highlight differentially expressed genes in relevant tissue and in particular on SBDR genes analyzed in this work (Appendix A). A total of 99 differentially expressed SBDR transcripts (FDR < 0.05 and absolute fold change of at least 40%) were identified. The highest number of SBDR transcripts are present in the skin (not exposed to the sun), with 25 SBDR genes, followed by the intestine (17), whole blood (14), lever (13), lungs (5), and kidney (2) (Figure 2B). Among all the genes considered in this analysis, 17% belong to the VIP class. Concerning the classification of DrugBank, 60% of the genes are drug targets, 33 are metabolizing enzymes, and 7% are drug transporters. No drug carriers were identified as differentially expressed (Figure 2C,D).

Additionally, in several genes—e.g., CYP3A7 (gene ID ENSG00000160870), CYP1A1 (gene ID ENSG00000140465) in liver, PLA2G2A (gene ID ENSG00000188257) in kidney—different transcripts with different functions are regulated in a similar way (Appendix A). This suggests a similar transcriptional regulation for all the transcripts and not the implication of posttranscriptional events such as degradation of specific RNA.

### 3.3. SBDR Genes in Liver

The liver is the most relevant site for drug metabolism. In this analysis, 17 transcripts were identified as differentially expressed: 12 are upregulated and 5 are downregulated in females as compared with males (Figure 3A,B and Appendix A). Of the analyzed genes, only 2 are VIP the CYP2B6 and CYP3A5, important members of the cytochrome P450 family. The highest upregulation (FC = 4.2, p.adj = 6 × 10^−06^) was observed for a protein-coding transcript encoding a non-canonical isoform of the cytochrome P450, CYP2B6. Two other P450 cytochromes upregulated in females were the CYP3A5 and CYP3A7. The differential expression can be observed for 1 transcript encoding a minor splicing isoform of the CYP3A5 (144 amino acids long) and for the four different transcript isoforms of the CYP3A7, three of which are non-coding transcript, and one is a gene/transcript containing an open reading frame for the CYP3A7 (ENST00000336374, FC = 2.3, p.adj = 3 × 10^−07^). An opposite expression pattern, downregulation in female, is observed for cytochrome CYP1A1, a cytochrome P450 monooxygenase involved in the metabolism of various endogenous substrates, including fatty acids, steroid hormones, and vitamins [28]. Three different transcripts were identified by the analysis, 1 is implicated in non-sense mediated decay (NMD) and 2 are isoforms encoding for canonical protein.

Finally, in the liver, 5 SBDR genes upregulated and 2 SBDR downregulated in females with a single transcript (Figure 3A,B) were identified. Particular mention should be made of an alternative isoform of the Exportin-1, XPO1 (ENST00000404992), which encodes the canonical protein. XPO1 mediates the nuclear export of cellular proteins and is a therapeutic target in many tumor types [29,30]. Then, one single differentially expressed transcript was identified in the following genes: the X-inactivation escaping-gene STS, transferrin receptor (TFRC), aldo-keto reductase family 1 member C2 (AKR1C2) and, a non-coding transcript of the Multidrug resistance-associated protein 1 (ABCC2) (Figure 3A,B). The genes downregulated in the liver are a transcript encoding the canonical isoform of iodothyronine deiodinase 3 (DIO3), and a protein-coding transcript, encoding the canonical isoform of the parathyroid hormone 2 receptor (PTP), a specific receptor for parathyroid hormone (Figure 3A,B).

### 3.4. SBDR Genes in Other Key Organs Implicated in Drug Metabolism

In the kidney only two SBDR genes with a single transcript were identified: the phospholipase A2 group IIA membrane enzyme (PLA2G2A) and the solute carrier family 2 member 9 (SLC2A9) (Figure 4A). The PLA2G2A, involved in inflammation and tissue [31,32], is a membrane enzyme with a single transcript upregulated in females. By contrast, the SLC2A9 gene, which has urate and fructose transmembrane transporter activity, is upregulated in males (Appendix A).

In the small intestine, 20 transcripts of 13 SBDR genes and the absence of VIP genes were found. From a functional point of view, the majority of SBDR are genes encoding drug targets. (Appendix A and Figure 4B). Interestingly, only one SBDR was downregulated in females (membrane spanning 4-domains A2 gene, MS4A2), while all the other genes (19 SBDR genes) were upregulated. Notably, MS4A2 gene product is one of the two targets of Omalizumab, a subcutaneous injectable controlling moderate-to-severe allergic asthma. Furthermore, the majority of the transcripts recognized by the analysis are encoded by 3 genes involved in lipid biosynthesis: diacylglycerol O-acyltransferase 2 (DGAT2), fatty acid desaturase 2 (FADS2), and fatty acid synthase (FASN). For example, 5 transcripts encoded by fatty acid synthase are coherently upregulated; two of them are protein-coding transcripts, of which are classified as “retained intron transcript” and one as “NMD transcript”. Fatty acid synthase is one of the targets of Orlistat, a reversible lipase inhibitor used in the treatment of obesity that works by inhibiting fat-metabolizing enzymes (according to Drugbank).

In the lungs, the analysis resulted in 6 transcripts from 6 different SBDR genes, of which 3 were upregulated and 3 downregulated in females. Specifically, 4 genes were defined as targets: 2 were key enzymes implicated in drug metabolism and 2 were defined as both target and transporter by DrugBank and no VIP genes (Appendix A and Figure 4C).

In whole blood, 20 transcripts from 14 SBDR genes were all upregulated in females as compared to males. Two of the identified genes were VIP: the CYP3A4 and the NAD(P)H quinone dehydrogenase 1 (NQO1). NQO1 is a detoxification enzyme that catalyzes the reduction of several substrates, such as quinones, alterations in NQO1 levels may lead to resistance to drugs including chemotherapeutics. Furthermore, 13 genes are drug targets (according to DrugBank), 1 is a drug transporter and 9 are metabolizing enzymes (Appendix A and Figure 4D). Of the 20 transcripts, 8 are members of the aldo/keto reductase superfamily which are critical for drug metabolism and toxin detoxification in the human body [33] (Appendix A and Figure 4D).

Finally, in the skin, 41 SBDR transcripts corresponding to 16 different genes were identified, of which 30 were upregulated and 11 downregulated (Appendix A and Figure 4E). Among them, 4 key genes for the pharmacogenetic (VIP) were highlighted: alcohol dehydrogenase 1B (class I), beta polypeptide (ADH1B), cytochrome P450 family 3 subfamily A member 5 (CYP3A5), and prostaglandin I2 synthase (PTGIS). ADH1B and PTGIS are drug targets, while CYP3A5 is a key enzyme implicated in drug metabolism. A single transporter shows a reduction of expression (40% lower) in females, the transferrin (FC = 0.59). Furthermore, 28 transcripts are drug targets and 11 are enzymes, according to DrugBank classification.

## 4. Discussion

Improving our understanding of sex differences in medicine is critical for the comprehension of human physiopathology and for developing new strategies for precision medicine. Previous works have already analyzed sexually dimorphic gene expression patterns that can be potentially applied with key indications for therapy [27,34,35,36]. In this study, we focused on the analysis of SBDR transcripts, transporters, carriers, enzymes, and targets relevant for ADMETox according to DrugBank [26]. We also identified VIP genes in the generated data. SBDR gene expression was analyzed in all the 44 tissues present in GTEX. Subsequently, we focused the analysis on the most relevant tissues implicated in ADMETox, liver, kidney, small intestine, skin, and whole blood. We identified sex-biased drug response transcripts in 41 of the 44 analyzed tissues. We decided to focus on sex effect higher than 40% (average FC = >1.40 and <0.6). The identified genes have specific biological functions, as they are mainly classifiable as enzymes and transporters, which often also constitute pharmacological targets.

Among the enzymes, we confirmed previous findings on the gender specific differential expression of transcripts encoded by the cytochromes P450 (CYP) family genes in the liver, skin, and whole blood (Appendix A and Figure 3 and Figure 4). In our data, CYP2B6, CYP3A7, and CYP3A5 were upregulated and CYP1A1 was downregulated in females as compared with males in the liver, while CYP3A4 and CYP3A5 were upregulated in the skin and whole blood, respectively. All the CYP identified except CYP1A1 were classified as VIP. CYP3A enzymes are the most abundantly expressed P450 enzymes in the liver and were responsible for the metabolism of more than 50% of all clinically used drugs [37], while CYP2B6 made up approximately 2–10% of total hepatic CYP content [38]. Consistent with previous work [39], we observed a differential expression of CYP3A4 transcripts in the liver that is not significant after Benjamini–Hochberg correction (FC > 1.9; p.adj = 0.06). It has been shown that CYP3A4-substrate drugs such as antipyrine, alfentanil, erythromycin, midazolam, and verapamil [40] have a higher clearance in females, which persists even after adjustments for physiological factors (e.g., body weight). Analyses of CYP3A4 in the human liver have indeed shown ~2-fold higher levels of protein in females compared to male liver tissue [39]. By contrast, we discovered a significant difference in the regulation of CYP3A4 in the skin (FC = 3.16; p.adj = 0.01).

Furthermore, the expression of CYP2B6 is upregulated in the female liver as previously demonstrated by Lamba and colleagues [41] (Appendix A). The sexually dimorphic expression of P450s is mainly regulated by the plasma growth hormone (GH) release by the pituitary gland, which is characterized by significant sex differences [42]. In turn, these differences affect the pharmacokinetics and pharmacodynamics of several drug treatments, contributing to specific inter-individual differences in drug efficacy and toxicity.

We also observed upregulation of the sex-linked gene STS transcript, encoding the enzyme steroid sulfatase in liver (FC = 1.62, p.adj = 5.4 × 10^−10^), lungs (FC = 1.43; p.adj = 1.08 × 10^–36^), and skin (FC = 1.55, p.adj = 2.47 × 10^−29^). STS is located on the distal short arm of the X chromosome (Xp22.3), very close to PAR, and it escapes X inactivation [43]. Earlier studies demonstrated that the enzymatic activity of the STS is also higher in females than males [44], being also regulated by sexual hormones [45]. STS catalyzes the hydrolysis of various 3 beta-hydroxysteroid sulfates including neuroactive steroids; thus, sex difference in steroid sulfatase activity could explain why males and females are differentially vulnerable to disorders of attention and impulse control [46].

Other interesting examples of transcripts differentially expressed are the protein-coding transcript for the aldo-keto reductase 1C (AKR1C) and the transferrin receptor (TFRC). AKR1C2 and AKR1C1 are particularly active in catalyzing the reduction of endogenous and xenobiotic aldehydes [33,47]. AKR1C2 is upregulated in females both in the liver and in the skin, while AKR1C1 is upregulated in females only in the skin. The transferrin receptor plays an important role in iron homeostasis in cells and is classified as a drug target and transporter according to DrugBank. Upregulation of human TFCR in females has already been demonstrated in humans [48].

There is considerable evidence for sex-based differences in clinical and pre-clinical studies and, the consciousness of the relevance of these differences in response to drugs is extremely relevant. Furthermore, sex differences in the incidence of ADR have drawn significant attention. Sex differences in genes implicated in ADMEtox mechanisms are associated with the therapeutic effects and risk effects of medications [4]. Indeed, females have—1.5-fold greater risk than males for developing ADR [4,49]. Additionally, the associations of endogenous and exogenous sex hormones with specific disease gene expression contribute to sex differences in therapeutic response [4].

In our data, significant sex differences in the expression of 99 transcripts of 59 key pharmacogenes were identified, and some of them are described above in detail.

It should be noted that our analysis is based only on transcripts and as all transcriptomic analysis need to be properly considered. Indeed, it is well known that there is not a perfect correlation between mRNA expression and the abundance of the encoded protein. Modern approaches, for RNA and protein analysis, clearly demonstrated that transcript levels and cognate protein levels do not necessarily correlate due to regulation of translation and posttranscriptional event and that only 40% of the variability in protein levels can be explained by mRNA levels [50].

Overall, these results show that there is a clear sex difference in the expression of highly relevant pharmacogenes in key tissues involved in drug response. Furthermore, with the increasing accessibility to the transcriptomic datasets, the number of SBDR genes is likely to expand and of course, become more robust from a statistical point of view. Additionally, although some limitations exist in the current identified SBDR genes—sex differences are tissue- and parameter-specific [51,52]—the analyses overall provided several biological implications related to sex differences in human drug metabolism.

The resulting knowledge, together with the growing understanding of the effects of human variability [25], will allow a further step towards sex-specific and personalized therapies. Thus, the full individual’s genetic and genomic peculiarities need to be taken into account when determining the right therapy and the right dose of the drug.

## Figures and Tables

**Figure 1 biomolecules-11-01206-f001:**
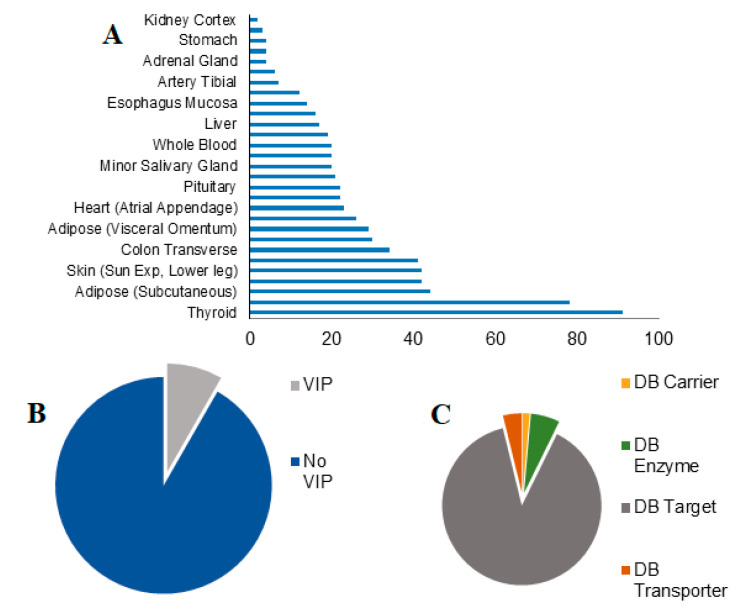
Sex-differential gene expression in pharmacogenes. (**A**) The number of sex-biased drug response genes (SBDR) identified per tissue (FDR < 0.05) are indicated. (**B**) Proportions of VIP and no VIP genes identified according to the PharmGKB classification. (**C**) Proportions of drug target, transporter, carrier, and enzymes identified according to DrugBank classification.

**Figure 2 biomolecules-11-01206-f002:**
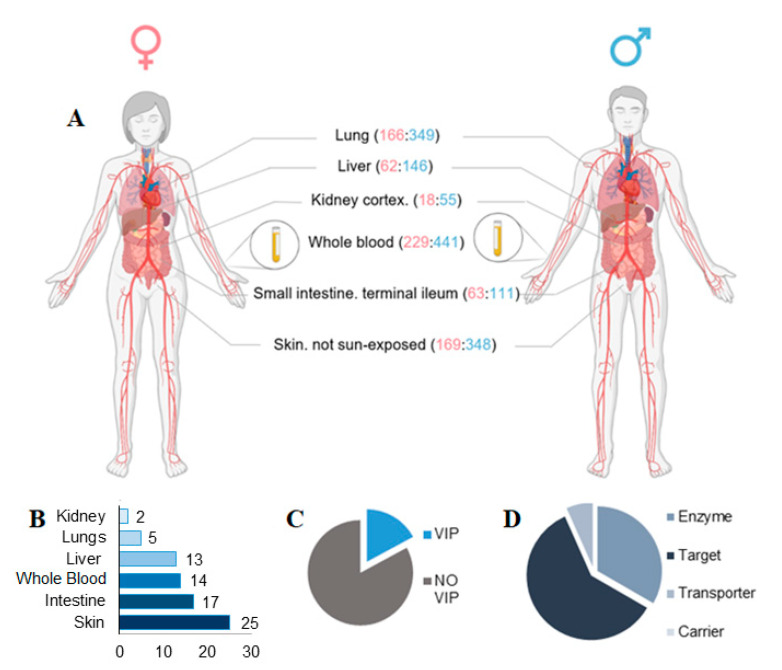
Sex-biases pharmacogenes identified in key tissue implicate in drug metabolism. (**A**) Tissue types relevant for drug metabolism are indicated, with sample numbers from GTEx v8 genotyped donors (females:males, in parentheses). (**B**) The number of SBDR identified in each tissue relevant for drug metabolism is indicated (FDR < 0.05). (**C**) Proportions of VIP genes and (**D**) drug target, transporter, carrier, and enzymes identified according to PharmGKB and DrugBank classification are indicate respectively. Panel A is created with BioRender.com.

**Figure 3 biomolecules-11-01206-f003:**
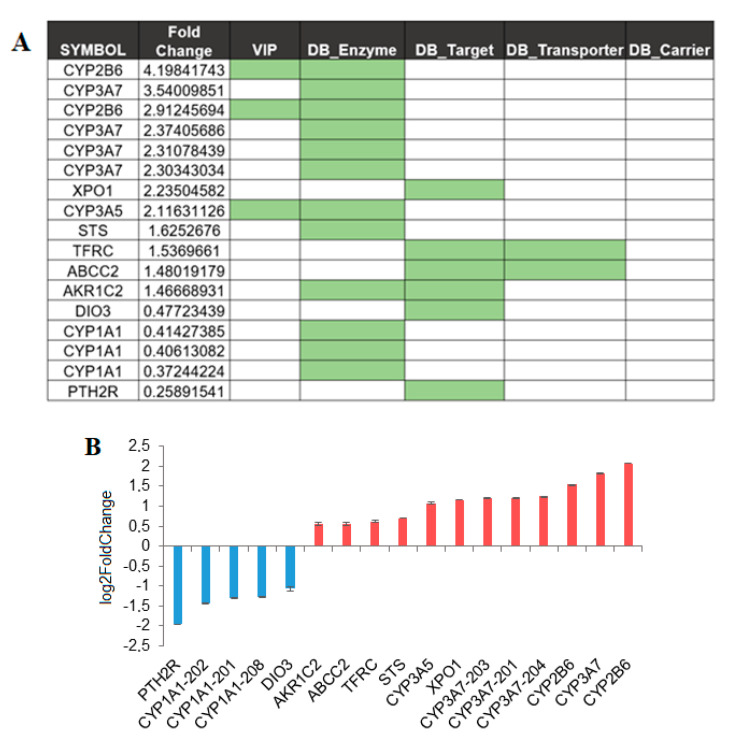
Sex affects gene expression in the liver. (**A**) SBDR transcript identified in liver, fold changes in female as compared to male are indicated. Transcripts that belong to one of the classes analyzed in this work, VIP and drug target, transporter, carrier, and enzymes are highlighted. (**B**) Transcripts showing differential abundance, which is at least 40% of up- or downregulation in females compared to males, were plotted.

**Figure 4 biomolecules-11-01206-f004:**
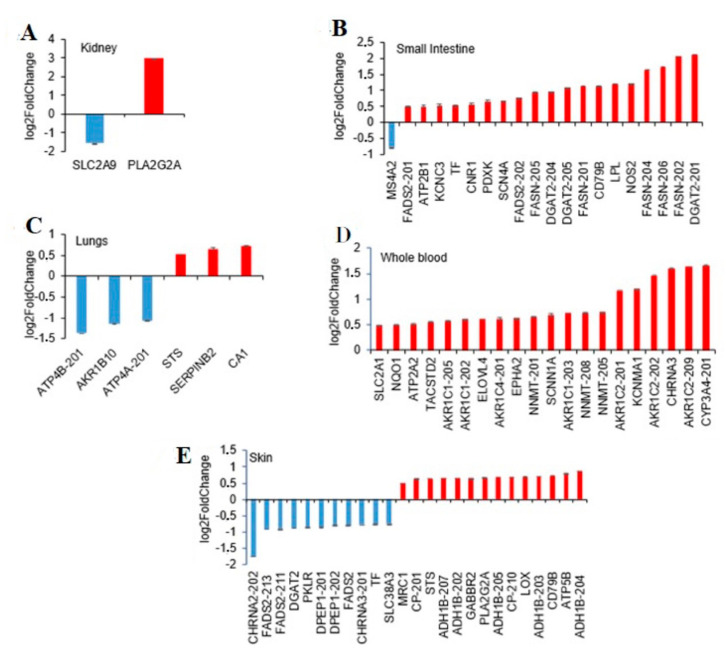
Sex-differential gene expression of pharmacogenes in relevant tissue for drug metabolism. Transcripts showing differential abundance, which are at least 40% of up- or downregulated in females compared to males, were plotted for the most relevant tissue implicated in drug metabolism. (**A**) Kidney. (**B**) Small intestine, terminal ileum. (**C**) Lungs. (**D**) Whole blood. (**E**) Skin, not exposed sun.

**Table 1 biomolecules-11-01206-t001:** The main characteristics of the dataset analyzed within this study.

Tissue	# Transcripts	# PKG-T	# of (%) Male	# of (%) Female	Mean Age
Liver	208	24	146 (70.20%)	62 (29.80%)	54.25
Lung	515	27	349 (67.76%)	166 (32.24%)	53.31
Kidney Cortex	73	4	55 (75.34%)	18 (24.66%)	56.28
Small Intestine	174	37	111 (63.80%)	63 (36.20%)	48.12
Skin	517	397	348 (67.32%)	169 (32.68%)	52.7
Whole Blood	670	54	441 (65.82%)	229 (34.18%)	51.82

Abbreviations: PKG-T, pharmacogenes encoded transcripts; #: number.

**Table 2 biomolecules-11-01206-t002:** Criteria for pharmacogenes transcripts inclusion.

Category	Source	Description
VIP	PharmGKB	Genes involved in metabolism and response to drugs. Often, VIP either play a role in the metabolism of many drugs or contain genetic variants which may contribute to severe drug responses.
Targets	DrugBank	Protein targets of drug action.
Enzymes	DrugBank	Proteins that are inhibited/induced or involved in drug metabolism.
Carriers	DrugBank	Endogenous proteins which bind to drugs and modify their pharmacokinetics and may facilitate transport in the bloodstream or across cell membranes (an example is albumin).
Transporters	DrugBank	Endogenous, membrane-bound, protein-based structure that physically moves drugs across cell membranes between the two sides of the cell membrane.

Abbreviations: VIP, very important pharmacogenes.

## Data Availability

https://gtexportal.org/home/datasets (accessed on 13 August 2021)https://go.drugbank.com/releases/latest#protein-identifiers (accessed on 13 August 2021)https://www.pharmgkb.org/downloads (accessed on 13 August 2021). https://gtexportal.org/home/datasets (accessed on 13 August 2021) https://go.drugbank.com/releases/latest#protein-identifiers (accessed on 13 August 2021) https://www.pharmgkb.org/downloads (accessed on 13 August 2021).

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
