# Peer review of "Sex-Biased Expression of Pharmacogenes across Human Tissues"

_biomolecules, 2021, doi:10.3390/biom11081206_

Round 1

Reviewer 1 Report

The authors present a well written, well referenced study of sex differences in critical endo- and xenobiotic metabolizing, transport and transformation proteins. This is a secondary analysis of an existing dataset. The statistical analyses and technical details are well reported. The results are clear and the graphical representations appropriate. The conclusions are supported by the data. I have the following comments, concerns and compliments:

Despite that this is an existing dataset, these are human tissues from the Gtex portal. This reviewer appreciates the URL supplied and that one can access the portal, but in the methods section a brief 1-2 sentence statement about which subset of data was used in the study and the consent/availability of the data (e.g. all are deidentified and anonymized) would be best practice.

The authors commonly report differences around the 2-fold level, and also “significant” differences below 2-fold different in transcripts. Anything 2-fold and above (similarly -2 fold or 50% loss plus) is the generally acceptable threshold, but it would strengthen the paper if the authors gave a rationale for assigning significant differences at e.g. 1.6-fold. Did they use a purely statistical approach, or were there other reasons for reporting less than 2-fold differences.

The authors should be circumspect re: assigning novelty in their finding of sex differences in CYP3A4, 3A5, 3A7 and 2B6. These have been reported multiply through mRNA, protein and clinical biomarker studies. Several of their other findings are, however; very novel.

The discussion would be strengthened with a brief consideration of transcript/protein discrepancies. This is particularly important for some of the transporters. Whilst it has been shown time and again in the literature the RNA transcripts are very good markers for CYP proteins, we no longer make that assumption about other enzyme or transporter families due to the effects of post transcriptional and post translational modification. This is not a fatal flaw, but should be clearly stated in the study as a caveat going forward.

Author Response

Reviewer 1

The authors present a well written, well referenced study of sex differences in critical endo- and xenobiotic metabolizing, transport and transformation proteins. This is a secondary analysis of an existing dataset. The statistical analyses and technical details are well reported. The results are clear and the graphical representations appropriate. The conclusions are supported by the data. I have the following comments, concerns and compliments:

[AU] We appreciate the reviewer’s positive remarks and helpful suggestions.

Despite that this is an existing dataset, these are human tissues from the Gtex portal. This reviewer appreciates the URL supplied and that one can access the portal, but in the methods section a brief 1-2 sentence statement about which subset of data was used in the study and the consent/availability of the data (e.g. all are deidentified and anonymized) would be best practice.

[AU] We thank the reviewer her/his suggestion. We added a new subsection “Data sources” in the Materials and Methods section, where all the details about the downloaded data are provided (Line: 153-156).

The authors commonly report differences around the 2-fold level, and also “significant” differences below 2-fold different in transcripts. Anything 2-fold and above (similarly -2 fold or 50% loss plus) is the generally acceptable threshold, but it would strengthen the paper if the authors gave a rationale for assigning significant differences at e.g. 1.6-fold. Did they use a purely statistical approach, or were there other reasons for reporting less than 2-fold differences.

[AU] In general RNAseq data suffers from several normalization and significance problems such as “arbitrary” fold change (FC) cut-offs and significance p-values of <0.05. Here we used an FDR of 0.05 and FC cut-offs of 40%. Of course the biological significance of a given FC is likely to depend on the gene and on the experimental context. On the other hand, it is realistic to assume that there is a minimum FC threshold below which differential expression is unlikely to be relevant for any gene. Here, in line with previous works [Oliva et.al], we found widespread sex-biased gene expression in all tissues, with 37% of genes exhibiting sex bias in at least one tissue, but with overall small (median FC = 1.04) sex effects when analysing PKG. For this reason, we decided to choose a FC smaller than the classical 2.

The authors should be circumspect re: assigning novelty in their finding of sex differences in CYP3A4, 3A5, 3A7 and 2B6. These have been reported multiply through mRNA, protein and clinical biomarker studies. Several of their other findings are, however; very novel.

[AU] We thank the anonymous reviewer for the suggestion to reshape some statements. We changed the sentence on line 22 and line 565-566.

The discussion would be strengthened with a brief consideration of transcript/protein discrepancies. This is particularly important for some of the transporters. Whilst it has been shown time and again in the literature the RNA transcripts are very good markers for CYP proteins, we no longer make that assumption about other enzyme or transporter families due to the effects of post transcriptional and post translational modification. This is not a fatal flaw, but should be clearly stated in the study as a caveat going forward.

[AU] We greatly appreciate the reviewer's suggestion. We have added a paragraph highlighting the limitation of transcriptomic based analysis (Line: 385-391).

Reviewer 2 Report

The manuscript "Sex-Biased Expression of Pharmacogenes Across Human Tissues" by Idda and colleagues uses data from the Genotype-Tissue Expression project (GTEx, v8 release) for 6 different compartments (liver, lung, kidney, small intestine terminal ileum, whole blood, skin not sun-exposed) that are thought to be relevant for human pharmacology to determine differential gene expression between males and females. The overall goal of the study was to provide a catalogue of sex-specific pharmacogenes, i.e. genes that have some kind of pharmacological importance. While I am certainly convinced that it is relevant to study these pharmacogenes in both sexes and the overall finding that there are quite a few differences in gene expression for males and females in these candidates and tissues is interesting per se, I am not entirely convinced by the conclusions. For example, I completely agree that one should consider sex-specific dosing recommendations or overcome sex-bias in clinical studies, but the presented study only gives sex-specific gene expression data for healthy individuals. What would be interesting – at least as a case study: given any disease state (where a drug is prescribed), are the sex-specific differences in gene expression still valid (see comment 3 below)?

Below I give some detailed comments which I hope help to improve the manuscript. I will go through the sections one by one.

Just a general comment: my questions/suggestions below should not be understood as general critisism at the current study. I am mainly lacking additional information that would possibly make the article more relevant for people that are not directly from the field of human pharmacogenomics and/or people that actually work on drug design and recommendations for patients.

Introduction

1) lines 42-45: I think this part could be extended. ‘Females consume more drugs on average than men’ – why is that the case? Are females really ‘sicker’, just older, or just taking care of their health more than men, for example?

2) The example of zolpidem: could also be extended. Why was the FDA recommendation adapted for females? Was the 50% lower doses just because of the female metabolism, was the correction just because of weight differences, what are the specific ADR in this example? Were interactions with other drugs expected as well as, for example, the drug is prescribed in combination with other drugs just in females? I think with a bit of elaboration the authors could make their point better: there are already sex-adjusted dosage recommendations for certain drugs, but so far for only a few and several questions have to be considered to make informed recommendations.

3) lines 65-69: I see that it is interesting to study sex-specific expression in tissues that are relevant for pharmacokinetics. However, the immediate question that comes into my mind: how can you transfer the information from healthy individuals to patients that are actually take the drugs in the end? I would ask the authors to at least comment on the fact that patients are not healthy and therefore might react differently to drugs. Could this be studied in a defined data set with healthy and non-healthy donors, for example?

Materials and Methods

4) lines 105-110: It does not become completely clear what is different between the two approaches – Oliva et al. and the current study – and I would ask the authors to elaborate on: what exactly are the ‘few differences’, why both approaches are ‘valid’ and what the authors mean with ‘comparable results’. These points stay quite vague.

Results

5) lines 242-247: Just for clarification: the two phase II clinical trials on breast cancer probably only include females? So, could this actually be an example for making use of the upregulation in females the authors detected here as these drugs are probably developed for females exclusively? I am slightly lacking the connection of the finding and the phase II trials.

Author Response

Reviewer 2

 The manuscript "Sex-Biased Expression of Pharmacogenes Across Human Tissues" by Idda and colleagues uses data from the Genotype-Tissue Expression project (GTEx, v8 release) for 6 different compartments (liver, lung, kidney, small intestine terminal ileum, whole blood, skin not sun-exposed) that are thought to be relevant for human pharmacology to determine differential gene expression between males and females. The overall goal of the study was to provide a catalogue of sex-specific pharmacogenes, i.e. genes that have some kind of pharmacological importance. While I am certainly convinced that it is relevant to study these pharmacogenes in both sexes and the overall finding that there are quite a few differences in gene expression for males and females in these candidates and tissues is interesting per se, I am not entirely convinced by the conclusions. For example, I completely agree that one should consider sex-specific dosing recommendations or overcome sex-bias in clinical studies, but the presented study only gives sex-specific gene expression data for healthy individuals. What would be interesting – at least as a case study: given any disease state (where a drug is prescribed), are the sex-specific differences in gene expression still valid (see comment 3 below)?

Below I give some detailed comments which I hope help to improve the manuscript. I will go through the sections one by one.

Just a general comment: my questions/suggestions below should not be understood as general critisism at the current study. I am mainly lacking additional information that would possibly make the article more relevant for people that are not directly from the field of human pharmacogenomics and/or people that actually work on drug design and recommendations for patients.

[AU] We appreciate the reviewer’s positive remarks and helpful suggestions.

Introduction

1) lines 42-45: I think this part could be extended. ‘Females consume more drugs on average than men’ – why is that the case? Are females really ‘sicker’, just older, or just taking care of their health more than men, for example?

[AU] We thank the reviewer her/his suggestion. More details have been added in the introduction section (Line: 42-48).

2) The example of zolpidem: could also be extended. Why was the FDA recommendation adapted for females? Was the 50% lower doses just because of the female metabolism, was the correction just because of weight differences, what are the specific ADR in this example? Were interactions with other drugs expected as well as, for example, the drug is prescribed in combination with other drugs just in females? I think with a bit of elaboration the authors could make their point better: there are already sex-adjusted dosage recommendations for certain drugs, but so far for only a few and several questions have to be considered to make informed recommendations.

[AU] We thank the reviewer her/his suggestion. More details have been added in the introduction section (Line: 53-56).

3) lines 65-69: I see that it is interesting to study sex-specific expression in tissues that are relevant for pharmacokinetics. However, the immediate question that comes into my mind: how can you transfer the information from healthy individuals to patients that are actually take the drugs in the end? I would ask the authors to at least comment on the fact that patients are not healthy and therefore might react differently to drugs.

[AU] We thank the reviewer her/his interesting comment. Pharmacokinetics is influenced by several factors, including the individual's state of health, the physiological hormonal status or, for women, their particular periods of life (menopause, pregnancy, menstruation). For clarity, we added some examples to the introduction section (Line: 61-74).

Could this be studied in a defined data set with healthy and non-healthy donors, for example?

[AU] We thank the referee for her/his valuable comment. We totally agree, sex-specific gene expression in tissues that are relevant for pharmacokinetics can be studied in non-healthy donors, but this is not the main focus of this manuscript. For sure we will use the reviewer suggestion as a starting point for future experiments.

Materials and Methods

4) lines 105-110: It does not become completely clear what is different between the two approaches – Oliva et al. and the current study – and I would ask the authors to elaborate on: what exactly are the ‘few differences’, why both approaches are ‘valid’ and what the authors mean with ‘comparable results’. These points stay quite vague.

[AU] We thank the reviewer her/his suggestion. In the revised version of the manuscript, we have included a more detailed description of the methodological differences of this work with Oliva et al. We justified our choices, mainly driven by different study aims, and we specified why both analytical approaches are valid. A direct comparison of the two paper results show that the great majority of the differential expressed genes we have identified, were statistically significant in Oliva et al. also, making us confident about the robustness of the approach (Line 132-146).

Results

5) lines 242-247: Just for clarification: the two phase II clinical trials on breast cancer probably only include females? So, could this actually be an example for making use of the upregulation in females the authors detected here as these drugs are probably developed for females exclusively? I am slightly lacking the connection of the finding and the phase II trials.

[AU] We thank the reviewer her/his suggestion. Undeniably, the two clinical trials were misleading. We replaced that sentence with a simpler one, indicating that FASN is one of the targets of an obesity drug.

Round 2

Reviewer 2 Report

Dear authors,

Thank you for addressing my previous concerns.